# Sports Injuries in Basketball Players: A Systematic Review

**DOI:** 10.3390/life14070898

**Published:** 2024-07-19

**Authors:** Nikola Aksović, Saša Bubanj, Bojan Bjelica, Miodrag Kocić, Ljubiša Lilić, Milan Zelenović, Dušan Stanković, Filip Milanović, Lazar Pajović, Ilma Čaprić, Vladan Milić, Tatiana Dobrescu, Constantin Sufaru

**Affiliations:** 1Faculty of Sport and Physical Education, University of Priština–Kosovska Mitrovica, 38218 Leposavić, Serbia; nikola.aksovic@pr.ac.rs (N.A.); ljubisalilic4@gmail.com (L.L.); 2Faculty of Sport and Physical Education, University of Niš, 18000 Niš, Serbia; miodrag.kocic73@gmail.com (M.K.); dukislavujac@gmail.com (D.S.); 3Faculty of Physical Education and Sport, University of East Sarajevo, 71420 Pale, Bosnia and Herzegovina; bojan.bjelica@ffvis.ues.rs.ba (B.B.); milan.zelenovic@ffvis.ues.rs.ba (M.Z.); lazar.pajovic@yahoo.com (L.P.); 4University Children’s Hospital, 11000 Belgrade, Serbia; filip.milanovic@udk.bg.ac.rs; 5Department of Biomedical Science, State University of Novi Pazar, 36300 Novi Pazar, Serbia; icapric@np.ac.rs (I.Č.); vmilic@np.ac.rs (V.M.); 6Department of Physical Education and Sport Performance, Vasile Alecsandri University, 600115 Bacau, Romania; sufaruconstantin@ub.ro

**Keywords:** injuries, basketball, knee, ankle, ACL

## Abstract

(1) Background: The objective of this systematic review was to collect relevant data in the available contemporary studies about sports injuries of basketball players and explain differences in sports injuries relative to gender, location, sport, and position on the court; (2) Methods: The papers were searched digitally using PubMed, MEDLINE, ERIC, Google Scholar, and ScienceDirect databases, from 1990 to 2024; (3) Results: The most frequent severe injuries for both genders are knee and ankle injuries and the most frequent forms of injury are ankle sprain and ligament strain. The most frequent injuries occur during running and after contact with the ball. Shooting guards sustain the highest injury rate followed by centers and point guards, while guards have the highest rate of adductor muscle injury; and (4) Conclusions: Studies indicate that ankle and knee injuries are prevalent among basketball players, with ankle sprains being particularly prevalent. Knee injuries are more common in female basketball players, including ACL injuries. Various factors contribute to injuries, including the biomechanics of jumping, landing, sudden changes in direction, and the physical demands placed on the body during the game.

## 1. Introduction

Basketball is a highly complex and demanding game comprised various types of jumps, start acceleration, and sudden changes in movement direction [1], where the body of a basketball player is subjected to significant physical stress which, in the case of physical unpreparedness, may lead to sustaining injuries [2]. Non-game activities tend to show more load than game activities in NBA players [3]. Moreover, Weiss and associates [4] indicated that a moderate working load is optimal for reducing injuries in men’s professional basketball. The majority of sports injuries are orthopedic in nature, and they are an inseparable part of any sports activity of both professional and recreational nature. During the sports game, players jump, pivot, and run in forward and backward directions and change the running course multiple times during the game [5]. The number of injuries among basketball players has been constantly increasing [6]. Therefore, prevention is of high significance.

There is a vast number of studies in the area of sports injuries and, consequently, a high number of studies pointing to the research conducted [7,8,9,10,11,12,13]. As the training technology advanced and players started facing higher physical demands, the risk of injury started to increase. Higher physical demands were placed on young teams as well, and they are expected to perform better, which can lead to a higher injury rate. There is a high number of prospective studies of amateur and professional basketball players of both genders, and they suggest that ankle and knee joints are the anatomic locations where injuries most frequently occur [14,15,16]. The most common injuries in basketball are ankle injuries [17], and the leading position among these injuries is occupied by ankle sprains [18]. Padua and associates [19] highlighted that an ankle sprain is the most common injury in basketball, with ankle fractures being the next most frequent. They also noted that athletes who experience an ankle sprain are likely to suffer a recurrence, which can develop chronic instability [19]. Also, the risk of ankle sprains increases among female basketball players, especially during the game, resulting in players’ absence from games for up to three weeks [20].

Drakos and associates [21] observed that in the National Basketball Association (NBA) players’ injuries of lower extremities are more common than injuries of upper extremities. The most frequent types of injuries were injuries to the knee and ankle [21]. Ankles and knees are constantly subjected to physical load during the game, making up over (50%) of all injuries in basketball [16]. A systematic review and meta-analysis findings indicate that preventive measures can be effective in decreasing the likelihood of injuries to the lower limbs and ankle sprains; however, they are not effective for anterior cruciate ligament (ACL) injuries [22]. The absence of knee stability dynamics is responsible for an increase in knee injuries among male and female basketball players, and it needs to be emphasized that there is no precise and accurate way to identify the increased risk of ACL injury [23]. Yeh and associates [24] searched the databases and identified 129 meniscus injuries during the 21 NBA seasons with higher occurrence in players with higher body mass index (BMI) [24]. The number of ACL injuries increases at the beginning of the regular part of the season and is almost identical before and after halftime. Therefore, it is necessary to perform a detailed analysis to test the potential role of fatigue in the increase in the frequency of ACL injury during the game and season [25]. Moreover, a comprehensive systematic review needs to be conducted to provide insight into injury prevalence in basketball players.

The objective of this systematic review was to collect relevant data in the available contemporary studies about sports injuries of basketball players and explain differences in sports injuries relative to gender, location, sport, and position on the court.

## 2. Materials and Methods

The review of the papers was conducted adhering to the methodological guidelines aligned with the Preferred Reporting Items for Systematic Reviews and Meta-Analyses (PRISMA) standards [26].

### 2.1. Search Strategy

Existing databases were explored, a summary and translation of the collected literature were conducted, and the study findings were evaluated. The papers were searched digitally using PubMed, MEDLINE, ERIC, Google Scholar, and ScienceDirect databases, from 1990 to 2024. The search utilized the following terms, either separately or combined: basketball players, injuries, knee joint, ACL, and ankle joint. Reference lists of included articles were scanned to identify additional relevant studies.

All titles and abstracts were examined for potential inclusion in the systematic review. Additionally, the lists of prior and original research were evaluated. Relevant studies that met the inclusion criteria were obtained after a thorough investigation. The research strategy was modified and adapted to each database research where that was possible with the aim of increasing the sensitivity of this review paper. All disagreements were resolved by agreement between two researchers or with the help of a third. Two authors (N.A. and B.B.) independently reviewed and selected the searched papers. Selected papers were then cross identified by the same two authors. The final decision for included papers was made by a third author (S.C.). The database search was limited to peer-reviewed journal articles published in English.

### 2.2. Inclusion and Exclusion Criteria

To determine if a study should be included, three authors independently evaluated the inclusion and exclusion criteria. The inclusion criteria were as follows: (1) randomized and non-randomized control trials published in English; (2) amateur, professional, young, adolescent, senior, male, and female basketball players; and (3) studies that included sports injuries in basketball and basketball injuries compared with other sports. The exclusion criteria were as follows: (1) duplicates; (2) conference abstracts; (3) case reports; (4) review articles; and (5) non-healthy athletes. Extracted data from selected studies were the following: first author and year of publication; participants; localization of the injury; and key findings.

## 3. Results

A total of 486 articles were found through the database search, and an additional six articles were discovered through reference lists. After removing duplicates and screening articles based on title and abstract, 64 studies were left. Two researchers independently evaluated these 64 studies. After the final screening, 36 studies were included in the systematic review. More details on the selection process can be found in Figure 1.

### Characteristics of the Studies

Twelve studies [11,12,13,21,24,27,28,29,30,31,32,33] included male participants. Seven studies [19,31,34,35,36,37,38] included female participants. Eighteen studies [8,9,10,16,20,23,25,39,40,41,42,43,44,45,46,47,48] included both male and female participants.

The sample consisted of young basketball players, adolescents, high school students [9,16,19,23,27,32,33,35,41,44,46], professional basketball players [11,12], NBA and Woman National Basketball Association (WNBA) leagues [13,21,24,28,30,31,38,45], and basketball players of all ranks and ages [10,20,40,43].

Six studies [25,36,37,39,42,47] in addition to basketball players had a sample made up of other athletes. Most included studies (*n* = 32) examined knee and ankle injuries, one study by Kamandulis and associates examined muscle injuries [29], while the study by Rodas and associates examined ankle and muscle injuries [12]. One study examined differences in injury incidence according to the position on the court [49]. Overuse injuries were analyzed in three studies [20,31,47]. Wrist and finger injuries were examined in five studies [32,39,40,46]. Lower back/pelvis injuries were addressed in three studies [10,32,47]. Trunk injuries are examined in three studies as well [33,46,47]. Concerning prevention programs, five studies analyzed knee valgus [23,27,36,44,48]. Hamstring strengthening and jump direction showed an influence on injury prevention [9,29,34]. A systematic review of sports injuries in basketball players is shown in Table 1 (results on prevalence) and Table 2 (results on prevention programs).

## 4. Discussion

The primary purpose of this systematic review of the contemporary literature is to collect relevant data about the sports injuries in basketball players, as well as to determine if there are any distinctions in sports injuries based on gender, location, sport, and court position. The main findings of this study show that lower extremity injuries, the ankle and the knee, are the anatomical locations where injuries occur most often in both genders, and the most frequent form of injury are ankle sprains and ligament strain. Injury rates are the highest among shooting guards, compared to centers and point guards. The clinical application of this study is reflected in the fact that, based on the obtained results, therapists have clearer data on the frequency of injuries in basketball, so that they can prepare their programs in advance to rehabilitate injured body parts. Also, this study is very important for biomechanics, to find prevention methods that would contribute to reducing the occurrence of injuries in basketball. However, the obtained results should be interpreted with certain limitations. The authors of this study did not examine the differences in injury rates between NBA and European basketball. It would also be interesting to examine the frequency of injuries between European countries or to examine the frequency of injuries during the regular part of the season and playoff competitions, which is a recommendation for future researchers.

### 4.1. Injuries Relative to Gender

Upon insight into the conducted research, a difference in sports injuries was identified between male and female basketball players. Female basketball players showed a higher possibility of knee injury, and an almost four-times higher probability of injury to the ACL, whereas the risk of injury is identical for both genders [8]. Ito and associates [10] found that there are no statistically significant differences in ankle and foot injuries between males and females (M: 24.8%; F: 23.8%) and for lower back injuries (M: 11.8%; F: 11.4%). Obtained results suggest that there were differences in terms of knee injury (M: 41.7%; F: 50.4%) and injuries to upper extremities (M: 9.7%; F: 5.1%) in favor of male basketball players. Among knee injuries, ACL injury predominates: 22.1% for male basketball players and 45.9% for female basketball players. They are followed by injuries of the meniscus with 13.2% for male basketball players and 9.6% for female basketball players and jumper’s knee with 14.8% for male basketball players and 7.2% for female basketball players [10]. Opposite results were obtained in other studies [15,41]. These studies suggested that injury rates between male and female basketball players show no significant difference during the game or training. In both genders the injuries with the highest rates are ankle sprains (M: 32%; F: 31%), knee injuries (M: 10%; F: 20%), and hand–fingers (M: 9%; F: 8%) [15]. The highest rate of severe injuries for both genders were ankle and knee injuries. Severity is not relevant to gender, height, age, or number of games played [43]. Basketball players of both genders who had compromised balance had almost seven-times more cases of ankle sprains in comparison to players with good balance, which suggests that balance may serve both as a predictor and protective factor for ankle sprain [41]. The constitution of the locomotor apparatus of the basketball players and basketball game structure can explain the occurrence of the highest rate of injuries to the lower extremities.

Using video analysis in the sagittal plane, it was established that in ACL injury, female athletes show higher lateral trunk and knee abduction angles than ACL-injured male athletes [45]. 3D kinematic analysis (3D) results suggested that female athletes demonstrated a greater valgus angle than male athletes. Results also suggested gender-related differences in ankle inversion in the standing position, and it should be noted that no differences were observed in gender-related knee flexion extension [44]. Using 3D kinematics during drop vertical jump (DVJ), it was established that female athletes have greater knee valgus and maximal knee valgus in the landing phase compared to male athletes [23]. Kinetic, kinematic, and electromyographic analysis suggested that directional jumps have a positive impact on knee biomechanics while reactive jumps showed significant differences, indicating discrepancies between controlled laboratory tests and real athletic performance. Both directional and reactive jumps have a significant role in injury prevention [9]. High-velocity elastic-band training can also be efficient in injury prevention [29].

### 4.2. Injuries Relative to Location

A retrospective analysis of data from the National Electronic Injury Surveillance System of the US Consumer Product Safety Commission from 1997 to 2007 estimated that 4,128,852 pediatric basketball-related injuries were treated in emergency departments. The most common injury among amateur basketball players was a strain or sprain of the lower extremities, especially the ankle (30.3% and 23.8%, respectively). Children aged 5–10 years were more likely to injure the upper extremities and sustain traumatic brain injuries (TBIs) and fractures or dislocations, while adolescents aged 15–19 years were three-times more likely to injure the lower extremities. The annual average of 375.000 injuries decreases by the year, but we must make sure to increase the tendency [46].

Research results suggest that in 207 basketball players, there were 97 injuries during one year, and the highest percentage were injuries of the lower extremities (66%) with the highest proportion being knee injuries (45%) [47], while Padua and associates [19] showed that ankle sprain makes up the majority of basketball injuries (76.7%). Cumps and associates [20] indicated that ankle sprains and overused knee injuries are the most frequent basketball injuries, and they make up 14.8% of the total number of basketball injuries [21]. Similar results were reported by Pasanen and associates [16]. They found that 78% of the total number of basketball injuries are injuries of the lower extremities, where 48% are ankle injuries and 15% are knee injuries, while the majority of the injuries are joints and ligament injuries (67%). Authors also warned about the alarming number of repeated ankle sprains [16]. Relatively great height with the use of various types of movement in a small space using high velocity and explosive movements leads to stress on the two most commonly injured joints.

Similar results were confirmed for female basketball players. Walters [31] performed a study, observing WNBA female basketball players over the years, and stated that the most frequent injuries are knee injuries (15.2%) and ankle injuries (14.4%). Similar results were confirmed by McCarthy and associates [37]. In this particular study, the most frequent injuries are ankle sprain (47.8%) and hand injury (20.8%). Therefore, prevention programs focused on these injuries, especially acute knee injuries, are crucial and call for further research.

One interesting piece of data is that injuries to muscles and the ankle, respectively, comprise 21.2% and 11.9% of all injuries, and the muscle injury rate is 1.8-times greater than the ankle sprain rate [12]. As a shortcoming of this study, the authors also stated that muscle injuries were mostly observed in comparison to ankle sprains. Patel and associates [13] found that 18.5% of players suffered a repeated adductor muscle injury and that the injuries resulted in no significant changes in either of the larger statistical categories observed (points, assists, rebounds, steals, blocks, turnovers, and field goal percentage). The authors concluded that NBA players returned to gameplay after adductor injury, missing an average of 16 to 17 days or 7 to 8 games. The adductor muscle injury does not affect player performance, game availability, or career longevity [13]. Therefore, further studies in the area of muscle injury prevention strategy are required.

Studies previously performed also suggested that injuries to the lower limbs are the most prevailing, and that, in most cases, injured body parts are the knee and ankle, followed by hands [33,40,43]. Prebble and associates showed that 72% of injured players recovered after two weeks and that 23% [40] of injured players were unable to play for over 28 days [16]. Cumps and associates [20] found that 37.4% were injuries without contact with the opponent, and that, in acute knee injuries, the absence period was seven to nine weeks [20].

Jumping or landing caused the most injuries (28.1%) [33], with the knee joint being the most vulnerable. Injuries mostly occurred in the offensive court half, and cryotherapy was the most used treatment [33]. In the study of Abdollahi and Sheikhhoseini [32], the highest percentage of injuries were in the ankle area (26.9%), lower back (15.5%), and knee (15.7%). The key factor for decreasing ankle sprains, knee fractures, and reduction in back pain is proprioceptive control [11]. Surgery is required in the case of serious injuries, and the most frequent types are meniscus surgery and knee arthroscopy [38].

### 4.3. Basketball Injuries Relative to the Position on the Court

Contemporary studies suggest that the frequency of injury occurrence depends on the position on the court. Injury rates were highest among shooting guards (47.8%), centers (34.8%), and point guards (17.4%). Risk factors for shooting guards and centers include age, height, weight, and training duration, while among point guards, only weight was associated with injuries. Additionally, individual training characteristics may be linked to the risk factors for certain positions, while weight presents a risk factor to players in all positions. No statistically significant difference was found between male and female basketball players regarding their position on the court [48]. One limitation of the study was the sample size of 204 participants, relative to the position on the court, and the other was a retrospective factor.

Cumps and associates [20] concluded that knee injury is the most frequent in center players (26%), relative to guard players (20%) and forward players (12%), while a statistically significant difference was observed between forwards and center players [20]. Results of the study performed by Patel and associates [13] suggest that adductor muscle injuries occur mostly among guards (49%) relative to forwards (25%) and centers (25%), and all the players were able to return to play (RTP) after the first adductor injury. These findings suggest a high RTP rate after adductor injury in the National Basketball Association (NBA) players with no significant effect on the player’s performance [12], while in ACL injury, NBA players exhibit a high rate of return to play after ACL reconstruction [28]. Patel and associates [13] emphasized that the data should be interpreted considering limitations, mainly due to using public archives to identify adductor injuries. The authors suggest that they are unable to confirm the report’s accuracy, but they also emphasized that this method was previously confirmed in several trials [13]. Opposite results were obtained in the study performed by Walters [31]. The author emphasized that there was no statistically significant difference between guards, forwards, and centers in the ACL injury rates. It was also noted that guards and forwards in the NBA are likely to sustain knee injuries, whereas centers are more likely to sustain back injuries. Moreover, Tosarelli and associates [48] found a higher incidence of ACL injuries in guards than the other positions in basketball. However, the Chi-square analysis revealed that there is a statistically significant difference in the low back injury occurrence, with the highest incidence in centers, in comparison to guards and forwards. It was mentioned in the previous lines that severe injuries require surgery [31]. Shoulder and elbow injuries showed that returning from shoulder and elbow problems did not influence shooting accuracy, but decreased player efficiency rating after a dominant shoulder injury was observed [50]. Besides ACL and meniscus surgery, reported surgical interventions included knee arthroscopic surgery (11.7%), ankle ligament reconstruction (2.6%), and shoulder stabilization (2.0%). The authors concluded that there are no differences between rates of the meniscus and ACL surgeries relative to the position on the court, and the above-mentioned surgeries had no influence on the length of career in the Women’s National Basketball Association [38].

### 4.4. Basketball Injuries Relative to Other Sports

In handball and basketball, injuries most frequently occur while running and after contact with the ball, while in soccer, injuries occur in duels with the opponent. The number of injuries per 1000 playing hours was increased for soccer players (5.6) relative to handball players (4.1) and basketball players (3.0). Ankle sprains were 25% of the total number of injuries, and finger sprains were 32%. The most severe injuries and the longest rehabilitation period were observed in soccer players [39]. Leppanen and associates [47] showed that injuries to the lower limbs are the most frequent injuries in basketball (66%) and floorball (55%), and that injuries are mostly to the knee in basketball (45%), whereas in floorball those are lower back/pelvis (39%) and knee (34%).

Anderson and associates [25] showed that the lowest number of non-contact injuries was observed in soccer, relative to basketball or lacrosse. ACL injury rates rose early in the regular season, particularly among lacrosse players. Injury frequency remained consistent before and after halftime, suggesting a need for further analysis of injury timing within each halftime [25]. Interesting results were, however, obtained in the study performed by Munro and associates [37]. Results of this study suggest that ACL injury occurrence is greater in female football players, and that no differences were found between sports in the DVJ task. Basketball players demonstrated significantly greater frontal plane projection angle (FPPA) values upon single-leg land (SLL) than female football players. Sallis and associates [42] found gender-related differences only in sports such as water polo and swimming, while in the group of sports including basketball, cross-country running, soccer, tennis, and track, no statistically significant differences were observed between male and female athletes. The authors concluded that there are no recommendations for decreasing injury occurrence rates for athletes competing in these sports.

The results indicated an 81% reduction in ankle sprain and low back injury (77.8%) appearance from the first to the third biennium while the reduction in knee sprains was not significant. A high level of improvement in perceptive stability of technical skills and movement control leads to efficiency in injury prevention not only in basketball, but also in other sports as well [8], and injuries occur more often in the second part of the game, as noted in the study performed by Walters [31].

## 5. Conclusions

The main contribution of this study is showing that the ankle joint and the knee are the most frequently injured in both genders. The most frequent forms of injury are ankle sprains and ligament strain. Knee injuries are more common in female basketball players, including ACL injuries, compared to their male counterparts. Shooting guards sustain the highest injury rate, and they are followed by centers and finally point guards, while guards have the highest rate of adductor muscle injury, followed by forwards and centers. Basketball players experience fewer injuries compared to handball and soccer players, with ankle sprains and knee injuries being the most common.

Various factors contribute to injuries, including the biomechanics of jumping, landing, sudden changes in direction, and the physical demands placed on the body during the game. Prevention programs, such as prophylactic exercises and technique training, appear effective in lowering the risk of injuries, especially ankle sprains.

## Figures and Tables

**Figure 1 life-14-00898-f001:**
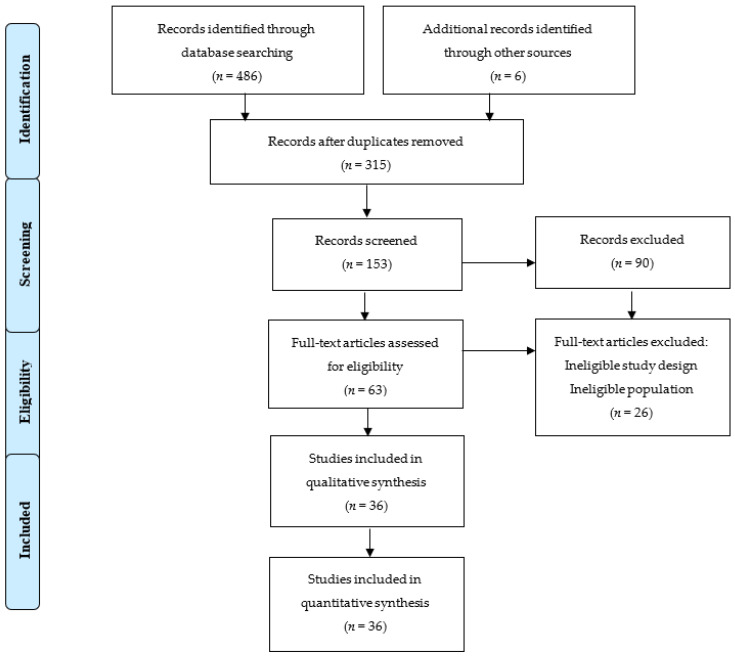
Flow chart diagram of the study selection.

**Table 1 life-14-00898-t001:** Prevalence of sport injuries in basketball.

First Author and Year	Participants	Localization	Key Findings
Yde and Nielsen (1990) [39]	Male and female adolescent athletes participating in three ball sports (soccer, handball, and basketball) *n* = 302	Knee Ankle Fingers	The injury rates (number of injuries per 1000 playing hours) were 5.6 for soccer, 4.1 for handball, and 3.0 for basketball. Ankle sprains represented 25% of the injuries, finger sprains 32%, thigh and leg strains 10%, and tendinitis/apophysitis 12%. The most severe injuries included four fractures, one ACL rupture, and two meniscus lesions. Soccer had the most severe injuries, requiring the longest rehabilitation periods. Tackling and contact with opponents were common causes of injury in soccer, while ball contact and running were frequent causes in handball and basketball.
Prebble et al. (1999) [40]	Male and female patients with sports-related injuries *n* > 6000	Ankle Fingers	A total of 19% were injured playing basketball. A total of 66.4% of the injured individuals were males, and the majority of injuries (53%) happened during school-related activities. A significant percentage (78%) of injuries occurred between the ages of 10 and 19. The most frequently injured body site was the ankle (33.1%), followed by finger injuries (19.3%), with sprains and strains accounting for the majority (55%) of injuries. The most common mechanism of injury (37.4%) involved no contact with other players. The vast majority of injuries (99%) were treated as outpatients. Around 72% of cases were expected to recover within a 2-week period.
Messina et al. (1999) [8]	Male and female students; high schools in Texas *n* = 100	Knee Ankle	Injury rates were similar between boys (0.56) and girls (0.49), with NS difference in the risk per hour of exposure. Sprains were the most common injuries for both, with the ankle and knee being the most commonly affected areas. Female athletes had a significantly higher rate of knee injuries, including a 3.79-times greater risk of ACL injuries. The risk of injury during games was significantly higher than during practice for both sexes.
McGuine et al. (2000) [41]	Male and female high school basketball players *n* = 210	Ankle	Subjects with ankle sprains scored 2.01 ± 0.32, while those without scored 1.74 ± 0.31. Higher postural sway indicated more ankle sprains. Poor balance correlated with nearly seven-times more ankle sprains than good balance.
McKay et al. (2001) [43]	Male and female basketball players *n* = 10.393	Knee Ankle	The overall injury rate was 18.3 per 1000 participations; 24.7 per 1.000 h. Serious injuries, missing a week or more, occurred at 2.89 per 1000, with the ankle being the most common (1.25), followed by the calf/leg (0.48) and knee (0.29). More severe injuries were linked to the lower limb, regardless of competition level, gender, age, height, games played, training, injury type, or injury mechanism.
Sallis et al. (2001) [42]	College athletes of both genders in seven similar sports (basketball, cross-country, soccer, swimming, tennis, track, and water polo) *n* = 3767	Back/Neck Shoulder Hip Thigh Knee Lower-leg Foot	Injuries were sustained by 45.7% of female athletes and 54.3% of male athletes. NS gender difference was found for injuries per 100 participant-years (52.5 for females vs. 47.5 for males). Significant differences were noted in swimming and water polo: female swimmers had more back/neck, shoulder, hip, knee, and foot injuries, and female water polo players had more shoulder injuries. Overall, female athletes reported higher rates of hip, lower-leg, and shoulder injuries, while male athletes had more thigh injuries.
Walters (2003) [31]	Female basketball players in WNBA *n* = 813	Knee Ankle	The knee (15.2%), ankle (14.3%), and patella (6.8%) were the most frequently injured body parts. Sprains (28.4%) were the most common injuries, with 49.4% affecting the ankle. Other injuries included tendonitis (19.6%), strains (18.6%), contusions (13.3%), and fractures (4.8%). NS difference in game-related injuries was found among guards, forwards, and centers. The highest injury incidence occurred during defensive rebounding (9.1%), offensive rebounding (6.0%), and driving (5.5%). Overuse/chronic injuries accounted for 20.2% of injuries. Injuries ending the season for the player made up 4.6% of all injuries, and 3.9% required surgery.
Cumps et al. (2007) [20]	Male and female senior players of all levels of play *n* = 164	Knee Ankle	The incidence of acute injuries was 6.0 per 1.000 h. Ankle sprains accounted for 20.7%, Overuse injury incidence was 3.8/1000. The knee incidence was 1.5/1000. The forward position experiences less knee overuse injuries compared to other positions. Overuse knee injuries and ankle sprains sprains accounted for >14.8%.
Randazzo et al. (2010) [46]	Male and female adolescent basketball players with injuries in the period 1997–2007 *n* = 4,128,852	Head Upper extremity Trunk Lower Extremity	Injuries occurred in the lower extremity (42.0%), upper extremity (37.2%), the head (16.4%), ankle (23.8%), and finger (20.2%). TBI injuries increased by 70%. Fractures or dislocations are higher in male athletes. TBIs and injury of the knee are higher in female athletes.
Drakos et al. (2010) [21]	Male basketball players in NBA *n* = 1094	Back Knee Ankle	Lateral ankle sprains accounted for 13.2% of injuries, patellofemoral inflammation accounted for 11.9%, lumbar strains accounted for 7.9%, and hamstring strains accounted for 3.3%.
Yeh et al. (2012) [24]	Male basketball players in NBA *n* = 129	Knee	Lateral meniscus accounted for 59.7% of injuries and the medial meniscus accounted for 40.3%. Injuries occured in the left and right knee equally. Medial meniscus (>30 years) Lateral meniscs (<30 years). BMI > 25 kg/m^2^ increased risk of meniscal tear. BMI < 25 kg/m^2^ decreased risk of meniscal tear. 19.4% players did not RTP.
Owoeye et al. (2012) [33]	Male and female adolescent basketball players *n* = 141	Upper extremity Trunk Lower Extremity	Incidence rate for male atlhetes was 1.1 injuries per match. Incidence rate for female athletes was 0.9 injuries per match. Jumping/landing accounted for 28.1% of injuries, lower extremities 75%, and knee 40.6%. Wrist and fingers, hip, and leg accounted for 3.1% and offensive half of the court accounted for 41%.
McCarthy et al. (2013) [38]	Female basketball players with injuries in the period 2000–2008 in WNBA *n* = 506	Head Shoulder Hand Knee Ankle	Ankle sprain accounted for 47.8% of injuries, hand injury 20.8%, patellar tendinitis 17.0%, ACL injury 15.0%, meniscus injury 10.5%, stress fracture 7.3%, and concussion 7.1%.
lei et al. (2013) [49]	Male and female adolescent basketball players *n* = 204	Upper extremity Lower Extremity	Injury incidence in shooting guards was 47.8%, injury incidence in centers was 34.8%, and injury incidence in point guards was 17.4%.
Ito et al. (2014) [10]	Male and female basketball players *n* = 1219	Upper extremity Lower back Knee Ankle Foot	The knee was the most often injured joint, with the foot and ankle, upper extremities, and lower back following closely behind. Female knee injury accounted for 50.4% of injuries, male knee injury accounted for 41.7% of injuries, female upper extremity injury was 5.1% of injuries, and male upper extremity injury was 9.7%. Most common was ACL injury. Least common was Osgood–Schlatter disease.
Leppanen et al. (2015) [47]	Male and female team sports athletes (basketball and floorball players) *n* = 401	Head/Neck Upper body Trunk Lower back Hip Thigh Knee	A total of 190 overuse injuries (47.4%); basketball injury incidence was 51%, lower extremities accounted for 66% of injuries, knee 45%, trunk 33%, lower back/pelvis 28%, shin/calf 11.4%, and groin 4%.
Minhas et al. (2016) [28]	Male basketball players in NBA *n* = 129	Hand/Wrist Knee Achilles tendon	The RTP rates for hand/wrist fractures was 98.1% and for achilles tears was 70.8%. Age ≥30 years and BMI ≥ 27 kg/m^2^ were predictors of not RTP. Achilles tendon rupture had a negative effect on career length and performance after recovery. Knee surgeries negatively affects performance after recovery.
Riva et al. (2016) [11]	Professional male basketball players *n* = 55	Low back Knee Ankle	↓ in the occurrence of ankle sprains (81%), low back pain ↓ (77.8%), and reduction in knee sprains (64.5%). Enhancements in single-stance proprioceptive control could be crucial for a successful decrease in low back pain, knee sprains, and ankle sprains.
Pasanen et al. (2017) [16]	Male and female adolescent basketball players *n* = 201	Knee Ankle	Injury incidence was 2.64 per 1000 h, and injury rate was 34.47 in basketball games and 1.51 in team practices. IRR between game and practice was 22.87. Lower limbs accounted for 78%, ankle 48%, knee 15%, and joint or ligaments 67%. NS differences were observed in injury rates between females and males during games and practices.
Anderson et al. (2019) [25]	Male and female sports athletes (basketball, lacrosse, and soccer) *n* = 529	Knee	Preseason IRR was 1.86, middle regular season IRR was 1.48, late regular season IRR was 1.56, and postseason IRR was 2.20. IRR of 2.18 indicates that female athletes had a greater injury rate than male athletes. Among all ACL injuries, 50% were in basketball players, 24% were in lacrosse athletes, and 26% were in soccer players. Early regular season before halftime IRR was 0.38 and after halftime in the late regular season the IRR was 2.40.
Rodas et al. (2019) [12]	Professional male basketball players *n* = 59		Muscle and ankle.
Patel et al. (2020) [13]	Male basketball players in NBA *n* = 65	Adductor	Guards accounted for 49% of injuries, forwards 25%, and centers 25%, and the adductor re-injury rate was 18.5%. Adductor injuries did not change any statistical parameter; an average of 16–17 days on the court are missed by NBA players after adductor injury.
Abdollahi and Sheikhhoseini (2022) [32]	Male basketball players (professional super league and first-divison league) *n* = 204	Ankle, Lower Back/Pelvis, Knee, Wrist/Fingers, Shin/Calf	Total of 628 injuries (6.07 injuries/1000 h). Acute ankle injuries accounted for 26.9% of injuries, lower back/pelvis injuries 15.5%, knee injuries 15.7%, wrist/fingers injuries 13.4%, and shin/calf injuries 14.2%. Mean time loss in first division league was 7.84/1000 h exposure, and mean time loss in professional super league was 4.30/1000 h exposure. Injuries during practice were more frequent than during competition.
Tosarelli et al. (2024) [48]	Male basketball players (professional European basketball leagues) *n* = 38	Knee (ACL)	Injuries while attacking accounted for 69% of injuries and injuries while defending 31%. Direct contact injuries accounted for 3%, indirect contact injuries 58%, and noncontact injuries 39%. Most injuries occurred during offensive cut, landing from a jump, and defensive cut. Most knee injuries occurred during sagittal plane flexion and valgus loading. More injuries were observed during the first ten minutes of a player’s effective playing time, notably in the scoring zone and among guards.

Legend: ACL—Anterior cruciate ligament; NBA—National Basketball Association; WNBA—Women’s National Basketball Association; BMI—Body mass index; TBIs—Traumatic brain injuries; RTP—Return to play; *n*—Number of participants; NS—Not statistically significant *p* > 0.05; *p* < 0.01; ↓—Statistically significant reduction *p* < 0.05; *p* < 0.01 ; ±—Mean and standard deviation; IRR—Incidence rate ratio.

**Table 2 life-14-00898-t002:** Prevention programs for basketball injuries.

First Author and Year	Participants	Localization	Key Findings
Ford et al. (2003) [23]	Male and female high school basketball players *n* = 81	Knee	KMA (3D) examined the valgus knee during DVJ performance; female athletes exhibited more total valgus knee motion and a larger maximum valgus knee angle than males. They also showed significant side-to-side differences in maximum valgus knee angle. Lack of dynamic knee stability, often not assessed before participation, may contribute to higher knee injury rates in females.
Ford et al. (2005) [44]	Male and female adolescent middle and high school basketball players *n* = 126	Knee Ankle	KMA (3D) examined knee valgus; females showed greater knee valgus angles compared to males. Gender differences also appeared in maximum ankle eversion and inversion during stance. NS differences were found in knee flexion angles. These variations in knee and ankle movements may explain higher ACL injury rates in females.
Sell et al. (2006) [9]	Male and female healthy high school basketball players *n* = 35	Knee	Jump direction had a major effect on ground reaction forces, joint angles, proximal anterior tibial shear forces and knee joint moments. Female participants demonstrated different KMA, KA, and EMG parameters during jump direction tasks. The direction of the jump greatly affected knee biomechanics.
Golden et al. (2009) [35]	Female collegiate basketball athletes *n* = 13	Knee	Internal rotation angle in knee was correlated with step width. Peak flexion, knee flexion, and internal rotation are associated with lateral false step. Lateral false step can increase injury risk of ACL.
Hewet et al. (2009) [45]	Male basketball players in NBA and female basketball players in WNBA *n* = 23	Knee	Injured female athletes demonstrated higher knee abduction and lateral trunk angles compared to male athletes and non injured athletes.
Wilderman et al. (2009) [34]	Female intramural basketball players *n* = 30	Knee	A 6-week agility program increased hamstring activation during ground contact. Agility training sessions can decrease injury incidence of ACL among female basketball players.
Koga et al. (2010) [36]	Female basketball and female handball players *n* = 10	Knee	Valgus loading in the knee indicates higher risk for ACL injury. Valgus motion occures 40 miliseconds after ground contact. Vertical ground-reaction force was 3.2 × body weight.
Munro et al. (2012) [37]	Female football players and female basketball players *n* = 93	Knee	Football and basketball female athletes had higher values for FPAA in SLL than in DJ. Basketball female players demonstrated higher FPPA values during SLL than football female players (ACL injury risk).
Paz et al. (2016) [27]	Young male basketball players *n* = 27	Knee	Knee valgus angle difference during the DVJ exercise was not found. During FSUP, a difference was observed between the non-dominant and dominant limbs.
Padua et al. (2019) [19]	Young female basketball players *n* = 28	Ankle	Right ankle dorsiflexion ↑ in EXP. NS improvement was reported in CON group. There was ↑ in left ankle in EXP group. EXP group ↑ ROM in right and left ankle and the COP. Single-leg stance barefoot with eyes closed, triceps sural stretching, and plank forearm position can decrease injuries in ankle area.
Kamandulis et al. (2020) [29]	College male basketball players *n* = 18	Rectus femoris Semitendinosus Biceps femoris	High-velocity elastic band training improved hamstring strength in male basketball players. High-velocity elastic band training can be used as a tool for injury prevention in hamstrings.
Morikawa et al. (2023) [50]	Male basketball players in NBA *n* = 126	Shoulder and elbow	Returning from shoulder and elbow problems did not influence shooting accuracy. Significant decline in player efficiency rating after dominant shoulder injury. Elbow or non-dominant shoulder injuries did not affect player efficiency rating. There is a correlation between younger age players and faster return to baseline player efficiency rating after shoulder injury.

Legend: ACL—Anterior cruciate ligament; NBA—National Basketball Association; WNBA—Women’s National Basketball Association; KMA—Kinematics; KA—Kinetics; EMG—Electromyography; DVJ—Drop vertical jump; FPPA—Frontal plane projection angle; SLL—Single-leg land; FSUP—Forward STEP-UP; ROM—Range of motion; COP—Center of pressure displacement; *n*—Number of participants; EXP—Experimental group; CON—Control group; NS—Not statistically significant *p* > 0.05; ↑—Statistically significant improvent *p* < 0.05; *p* < 0.01; *p* < 0.01.

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
