# Peer review of "Sports Injuries in Basketball Players: A Systematic Review"

_life, 2024, doi:10.3390/life14070898_

Round 1

Reviewer 1 Report

Comments and Suggestions for Authors

Congratulations for the paper. My comments are:

·       Abstract: I don't understand why the paper is about basketball and in the abstract it talks about football. The authors must modify it

·       Abstract: The conclusion should not be a reflection, but rather the finding of the study 

·       At the end of the introduction, the authors must state the objective of the study but must delete the last two sentences

·       the search strategy must be more complete

·       Table 1 should be designed differently. That's not legible

·       The discussion section should begin with the main finding of the review. 

·       Authors must include a clinical applications section in order to determine the practical purpose for therapists.

·       Authors should include a limitations section at the end of the discussion section.

·       The conclusion must be simpler and more concrete. They must write 2-3 lines with their main contribution

Comments on the Quality of English Language

Minor editing of English language required

Author Response

Dear,

Thank you very much for taking the time to review this manuscript and for contributing to the substantial improvement of our manuscript.

Please find the detailed responses attached and the corresponding revisions/corrections highlighted in the re-submitted file.

Kind regards

Reviewer 2 Report

Comments and Suggestions for Authors

l.34: Revise, this conclusion is not based on the results.

l.40: This paragraph needs 1-2 more sentences and more literature to explain the increased loads during game and training that may lead to injury.

l.55-83: This part is hard to follow, likely because this information contains many numbers and would fit better within the results section.

l.84: Here, there is the weakest point in the introduction. The ‘story’ is not tight as the gap that this review will fulfill has not been clearly identified.

l.88: I recommend a re-structure of the introduction: 1st par.: description of the sport and its loads leading in the end introducing the concept of injury; 2nd par.: what is known so far in basketball injuries (based mostly on previous reviews) about prevalence, sites of injuries and mechanisms; 3rd par: identify the gap; which points the previous reviews have missed? What new yours will add? Present clearly your aims.

l.89: Ethical approval?

l.105: Improve the style of writing in 2.2

l.117: Improve the style of writing in 2.3

l.156: What about European studies? About which countries?

l.162: The table is problematic as it presents information on different topics. I suggest splitting this table in separate tables each one addressing a separate question: e.g., one table for prevalence, another table for prevention programs etc.

l.345: Improve the style of writing in conclusions, that should be a single paragraph of 8-12 lines highlighting the most important information.

Author Response

(The authors gave the same response as above.)

Round 2

Reviewer 2 Report

Comments and Suggestions for Authors

I asked the editor to check the point 6 in the answer about my ethical concern.